# The Specificity of Determining the Latent Heat of Solidification of Cast Hypoeutectic AlSiCu Alloys Using the DSC Method

**DOI:** 10.3390/ma17174228

**Published:** 2024-08-27

**Authors:** Mile B. Đjurdjević, Vladimir Jovanović, Mirko Komatina, Srecko Stopic

**Affiliations:** 1Department of Material Science and Technology, University of Applied Science, Roseggerstrasse 15, 46000 Wels, Austria; djurdjevicmile506@gmail.com; 2Fuel & Combustion Laboratory, Faculty of Mechanical Engineering, University of Belgrade, Kraljice Marije 16, 11000 Belgrade, Serbia; vjovanovic@mas.bg.ac.rs; 3Department of Thermodynamics, Faculty of Mechanical Engineering, University of Belgrade, Kraljice Marije 16, 11000 Belgrade, Serbia; mkomatina@mas.bg.ac.rs; 4IME Process Metallurgy and Metal Recycling, RWTH Aachen University, Intzestrasse 3, 52056 Aachen, Germany

**Keywords:** latent heat of alloys, DSC, AlSiCu alloys, cooling rate

## Abstract

Latent heat is commonly measured using Differential Thermal Analysis (DTA) or Differential Scanning Calorimetry (DSC) or calculated using software packages (Thermo-Calc). In this study, the DSC method was used to comprehensively evaluate the accuracy of calculated latent heat for a specific range of cast AlSiCu alloys, considering their solidification under different cooling conditions. The tests involved varying concentrations of two crucial alloying elements: *w*_Si_ (5, 7, and 9%) and *w*_Cu_ (1, 2, and 4%). All selected alloys were analyzed under three distinct cooling/heating rates: 6, 10, and 50 °C/min. The Thermo-Calc method was used in this work to calculate the latent heats of the investigated alloys. The results obtained show good agreement between the measured and calculated values. The increase in silicon content in the investigated alloys from 4.85% to 9.85% resulted in the increase in latent heat from 407.6 kJ/kg to 467.5 kJ/kg. Higher cooling rates, such as 50 °C/min, resulted in a reduced latent heat release compared to slower rates such as 10 °C/min and 6 °C/min.

## 1. Introduction

Cast hypoeutectic AlSiCu alloys hold significant importance across various industries due to their unique combination of properties, making them ideal for a wide range of applications [1]. These alloys are highly valued for their superior mechanical properties, excellent castability, corrosion resistance, thermal conductivity, specific heat capacity machinability, and cost-effectiveness [1,2,3,4]. The main alloying elements, silicon and copper, are primarily responsible for defining the microstructures and mechanical properties of these alloys [5,6,7,8,9,10,11]. The castability and fluidity of these aluminum alloys have improved through silicon addition [12]. Additionally, the presence of silicon leads to reductions in shrinkage porosity, giving those alloys superior mechanical and physical properties [13,14]. Copper, as a second major alloying element, has been added to considerably increase the strength and hardness of AlSiCu alloys in as-cast and heat-treated conditions [15]. In addition, copper reduces the corrosion resistance of aluminum alloys and, in certain alloys, increases stress corrosion susceptibility [16]. This element is generally responsible for reducing the casting characteristics, especially the feeding ability, of AlSiCu alloys [17,18]. The versatility of hypoeutectic AlSiCu alloys makes them suitable for applications in the automotive, aerospace, electronics, and marine industries, underscoring their critical role in modern manufacturing and engineering. Understanding the value of latent heat for AlSiCu alloys is crucial for accurately simulating casting processes [19,20]. Latent heat plays a significant role in the solidification process, impacting the thermal behavior, microstructure formation, and overall quality of the cast product [20]. One of the primary reasons why knowledge of latent heat is essential in casting simulations is its influence on solidification and cooling rates [21]. The latent heat of an alloy directly affects the rate at which it solidifies and cools. These rates are critical as they affect the final microstructure and mechanical properties of the cast component [19,20,21]. The latent heat of solidification is the most important thermophysical property of a material. During the transformation from liquid to solid, each component of alloy releases a characteristic amount of latent heat. Latent heat is the energy absorbed or released by a substance during a phase transition, such as melting, vaporization, or sublimation, without a change in temperature. This energy is required to change the state of a substance without altering its temperature, facilitating transitions between solid, liquid, or gaseous phases. Latent heat is typically measured by Differential Thermal Analysis (DTA) or Differential Scanning Calorimetry (DSC) [22]. Computational thermodynamic software packages (e.g., JMatPro, FactSage, and Thermo-Calc) can also be used to calculate the latent heat of multicomponent systems. However, some authors note that their accuracy is constrained by the quality of the thermodynamic databases they rely on [23,24,25]. These software packages use the Scheil–Gulliver (SG) model to consider non-equilibrium solidification, assuming negligible solute diffusion in the solid state and complete diffusion in the liquid state [26]. This model is highly accurate for aluminum alloys and matches experimental results well [27]. Cooling curve analysis can also calculate the latent heat of cast aluminum alloys. This method measures temperature changes during the solidification of molten metal poured into a small thermal analysis cup [28]. It has been used in the casting industry for fundamental metallurgical studies and determining phase diagrams for hundreds of years [29]. The cooling curve method is favored in commercial applications for its simplicity, low cost, minimal sample preparation, consistency, and applicability on foundry floors. Numerous studies have employed TA techniques based on Newtonian, Fourier, and Energy Balance methods to calculate the latent heat of pure metals and alloys [7,30,31,32,33].

In this work, the DSC method was applied for the quantitative analysis of phase transitions, melting points, and enthalpy changes in investigated alloys [22]. This technique can be applied over a wide temperature range, making it versatile for studying various materials. DSC typically requires a small sample size (10–20 mg), which could be particularly significant in the case of inhomogeneous alloys, where the phases and intermetallic are not uniformly distributed. Accurate calibration is crucial for obtaining reliable results. Additionally, the precision of DSC results is severely influenced by the chosen heating and cooling rates. Having accurate latent heat data is crucial for the precise modeling of temperature gradients, predicting defects and controlling microstructure formation. It determines phase formation temperatures, influencing mechanical properties. Understanding latent heat aids in predicting dendrite formation and preventing common defects. Additionally, it helps model residual stresses, enabling process optimization for improved quality and efficiency. Consequently, knowledge of latent heat is fundamental for accurate casting simulation, enhancing product reliability and performance across various applications.

Our literature review provided no experimental data on the latent heat of investigated alloys. Considering the importance of these data, the authors focused on determining them using the most reliable tools, experimental (DSC analysis) and theoretical (Thermo-Calc method) techniques. In this paper, the results of performed investigations and calculations are presented.

The results from both methods are critically evaluated. This study aimed to comprehensively assess the accuracy of calculated latent heat for a specific range of cast AlSiCu alloys, considering their solidification under different cooling conditions. The focus of investigations was on the effects of varying concentrations of two crucial alloying elements—*w*_Si_ (5% to 9%) and *w*_Cu_ (1% to 4%)—on the amount of latent heat released during solidification. To analyze the impact of cooling rates on the released latent heat, three distinct cooling rates—6, 10, and 50 °C/min—were applied in each test. Besides the experimental tests, performed using Differential Scanning Calorimetry, the Thermo-Calc method was used to obtain theoretical results. Coupling the experimental with theoretical data will provide the method for the prediction of the latent heat of tested alloys.

## 2. Materials and Methods

The experiment included testing of selected range of AlSiCu alloys using Differential Scanning Calorimetry for three distinct cooling rates to determine released latent heat under these conditions.

### 2.1. Material

As listed in Table 1, nine synthetic AlSiCu alloys were produced by melting a charge of Al with 5% of Si, Al with 7% Si, and Al with 9% of Si base alloys and adding 1, 2, and 4% of Cu. The selected ranges of major alloying elements were chosen to cover the compositions of commonly used commercial aluminum alloys in various industries. The chemical compositions of the resulting alloys were determined using Optical Emission Spectroscopy (OES) in the following ranges: Fe (~0.1), Mg (~0.16), Zn (~0.01), Ti (~0.09), Sr (~0.0035), and Ni (~0.009).

### 2.2. Method: Differential Scanning Calorimetry

To obtain an unbiased microstructure and repeatable results, the DSC specimens were “rapidly quenched” by a massive copper chill and machined to a thickness of 0.5 mm and a diameter of 5 mm (average mass was ~20 mg) in accordance with the standard DSC procedure as detailed in the literature [34]. After the desired dimensions were reached, they were cleaned and placed into an alumina crucible. The DSC experiments were performed at the same heating and cooling rates of 6 °C/min, 10 °C/min, and 50 °C/min. Argon as a protective gas was used during the experiments. The DSC equipment was calibrated using a sapphire standard test sample. In Table 2, the thermal process cycles used for testing nine AlSiCu aluminum alloys are listed. Each thermal cycle included heating–maintaining–cooling process: (1) isothermal maintenance for 10 min to 25 °C; (2) heating to 800 °C with three heating rates, 6, 10, and 50 °C/min; (3) isothermal maintenance for 10 min to 800 °C; and (4) cooling to 25 °C with three cooling rates, 6, 10, and 50 °C/min. Each DSC experiment for each alloy was repeated two times and an average value for latent heat was taken for further consideration. The transformation energy for all tested DSC specimens was measured using a NETZSCH DSC 404 C Pegasus Differential Scanning Calorimeter operating between 25 and 1500 °C.

## 3. Results and Discussion

Differential Scanning Calorimetry (DSC) is a thermal analysis technique used to study the thermal behavior of materials, including phase transitions, melting points, melting ranges, enthalpy changes, latent heat, and specific heat capacity changes. For cast AlSiCu alloys, the DSC curve provides valuable insights into the melting, solidification, and phase transformation characteristics of the alloys. A typical heating and cooling DSC curve for cast AlSiCu alloys, presented in Figure 1, exhibits distinct features corresponding to these thermal events.

A typical DSC curve for cooling and heating cycles of the AlSi7Cu1 alloy scanned at a cooling rate of 6 °C/min is presented in Figure 1. On the DSC curves, the metallurgical reaction peaks reflect the specific phase changes and the peak area is proportional to the heat of reaction associated with the phase transformations. There are at least three distinguished peaks reacting in this type of AlSiCu-series aluminum alloys for both heating and cooling cycles. Peak 1 corresponds to the development of primary α-aluminum dendrites. Peak 2 represents mainly the appearance of the binary Al-Si eutectic phase. Peak 3 is associated with the formation of minor copper-rich eutectic phases. Some other thermodynamically weak events, such as the appearance of magnesium- and iron-rich phases, can also be recognized during heating and cooling, depending on the amounts of magnesium and iron present in the observed alloy.

In Figure 1, exothermic peaks are related to heating peaks 1, 2, and 3 (orange, pointing upward) while endothermic peaks are related to cooling peaks 1, 2, and 3 (blue, pointing downward). The heating cycle (orange) shows upward peaks, indicating heat is being released (exothermic). The cooling cycle (blue) shows downward peaks, indicating heat is being absorbed (endothermic).

### 3.1. Impact of Chemistry on Shapes of DSC Curves and Calculated Latent Heat

A review of the literature reveals a lack of data on the latent heat of solidification for AlSiCu-series aluminum alloys [20,21]. Additionally, there is significant variation in the calculated latent heat values for these alloys obtained through different analytical and measurement techniques. Therefore, in this paper, the impact of the chemistry of AlSiCu cast hypoeutectic alloys is analyzed using the DSC method.

The influence of an increasing silicon content on the shape of the complete DSC curve for heating and cooling cycles is illustrated in Figure 2 while Figure 3 shows the influence of an increasing copper content on the DSC melting and solidification paths.

The shapes of the DSC cooling curves are strongly influenced by the silicon and copper contents in the AlSiCu aluminum melt as presented in Figure 2 and Figure 3. Increasing the silicon content by a constant copper content significantly affects the onset and duration of the primary solidification of the α-Al dendrites. As illustrated in Figure 2, an increase in the silicon content from 5 to 9% shifts the liquidus temperatures to lower values by approximately 24 °C during both heating and cooling cycles. However, the addition of silicon, as depicted in Figure 2, does not significantly alter the nucleation temperature of the AlSi- and AlSiCu-rich eutectics. Additionally, it can be observed that an increase in silicon content decreases the primary solidification time of the α-Al dendrites. This indicates that silicon controls the fraction of solid in the primary and eutectic phases and influences the amount of latent heat released during solidification. As illustrated in Figure 3, the addition of copper, while maintaining a constant silicon content, also lowers the liquidus temperature by approximately 20 °C during both heating and cooling cycles. In contrast to silicon, the addition of copper slightly reduces the AlSi eutectic temperature by about 1.4 °C. Interestingly, copper increases the nucleation temperature of copper-rich eutectics by approximately 9 °C. Furthermore, at a higher copper content (4%), the appearance of two peaks in the DSC curves indicates the formation of two copper-rich phases with different structures and chemical compositions during solidification [35]. This suggests that copper significantly influences the variety and number of phases that crystallize during the solidification of AlSiCu alloys, thereby affecting the amount of latent heat released during the process. Generally, it can be concluded that the interaction between silicon and copper content affects the solidification process, phase formation, and, consequently, the latent heat released during the solidification of AlSiCu hypoeutectic alloys.

The alloy composition affects the amount of released latent heat as shown in Figure 4.

The latent heat values, read from DSC curves, increased from 407.6 kJ/kg to 467.5 kJ/kg as the silicon content increased from 4.85% to 9.85% in the investigated alloys. These results indicate that the effect of copper on the latent heat of the investigated alloys is not as significant as that of silicon. This was expected, given that the latent heat of pure silicon is approximately 1787 kJ/kg whereas the latent heat of pure copper is 206 kJ/kg [36]. As previously mentioned, increasing the silicon content in AlSiCu alloys lowers the liquidus temperature and shortens the primary solidification time of the α-Al dendrites [37]. This reduction in liquidus temperature and alteration in solidification dynamics can modify the total latent heat released. A higher silicon content typically increases the fraction of eutectic phases, which significantly increases the amount of released latent heat. The addition of copper also lowers the liquidus temperature but has distinct effects compared to silicon. As shown in Figure 3, copper addition results in a slight decrease in the AlSi eutectic temperature and a significant increase in the precipitation temperature of copper-rich eutectics. This leads to a smaller quantity of precipitated AlSi eutectics, resulting in the release of a smaller amount of latent heat.

### 3.2. Impacts of Various Cooling Rates on Shapes of DSC Curves and Calculated Latent Heats

The DSC curves presented in Figure 5 illustrates the thermal behavior of the AlSi7Cu4 alloy under different cooling rates. The graph plots the DSC signal (mW/mg) against the temperature (°C) for three distinct cooling rates: 50 °C/min, 10 °C/min, and 6 °C/min.

The solidification paths, as Figure 5 indicates, are notably different for each cooling rate. A high cooling rate (50 °C/min) leads to a quick solidification process, which can result in a finer microstructure due to limited time for the diffusion of solute elements, often resulting in smaller dendrites and possibly a higher degree of supersaturation. A moderate cooling rate (10 °C/min) creates a balance between fine and coarse microstructures, allowing some diffusion of solutes while maintaining a relatively fine structure. A low cooling rate (6 °C/min) allows for the extensive diffusion of solutes, resulting in coarser microstructures, with a more gradual solidification process that can lead to the formation of larger dendrites and potentially more segregation.

Based on the results from Figure 5, it can be expected that various cooling rates should have an impact on the amount of released latent heat. Figure 6 summarizes DSC measurements related to the latent heats released for nine investigated alloys.

As presented in Figure 6, higher cooling rates, such as 50 °C/min, result in a reduced latent heat release compared to slower rates such as 10 °C/min and 6 °C/min. Interestingly, the difference in released latent heat between the highest and lowest rates for each alloy is not as significant as might be expected. For instance, the highest difference was observed in the AlSi5Cu2 alloy (33.2 kJ/kg) while the lowest was 15.7 kJ/kg in the AlSi9Cu1 alloy. Clearly, the cooling rate does impact the released latent heat, although moderately. Several mechanisms can explain this behavior. Firstly, at higher cooling rates, there is insufficient time for the nucleation and growth of primary α-aluminum dendrites and secondary phases like AlSi- and copper-rich eutectics. This rapid cooling favors the formation of finer microstructures. In contrast, slower cooling rates (e.g., 10 °C/min or 6 °C/min) allow for the nucleation and growth of larger and more stable microstructures, which undergo more complete phase transformations, resulting in a greater release of latent heat. Additionally, higher cooling rates can induce undercooled solidification, where the alloy solidifies at a temperature below its equilibrium melting point. This undercooling limits the extent of phase transformation and reduces the latent heat released during solidification. Moreover, rapid cooling tends to trap some liquid phase within the solidified structure, further restricting the release of latent heat. The steeper thermal gradients associated with higher cooling rates can also lead to non-uniform cooling and localized differences in solidification, which can result in varying microstructures and further reduce the total latent heat released compared to materials cooled at slower rates. It can be assumed that slower cooling rates facilitate a more complete phase transformation from liquid to solid, leading to a higher latent heat release. In contrast, faster cooling rates hinder this transformation, resulting in a lower latent heat release. Therefore, while the cooling rate does impact the measured latent heat, its influence is not as significant as the alloy chemistry itself.

The measured (DSC) and calculated (Thermo-Calc 2024a, Database: TCAL9) values of latent heat for the nine AlSiCu aluminum alloys are summarized in Table 3. As shown in Table 3, the latent heat values for the selected cast AlSiCu alloys calculated using Thermo-Calc were mostly higher than those measured by the DSC method. This discrepancy could be attributed to several factors. Thermo-Calc relies on idealized models and thermodynamic data that assume perfect conditions, potentially oversimplifying real-world scenarios. Inaccurate phase diagrams and assumptions about perfect mixing can lead to higher calculated values. Conversely, DSC measurements face limitations such as calibration errors, heat losses, and issues with sample size and homogeneity, all of which can result in lower measured latent heat. Additionally, DSC experiments often operate under non-equilibrium conditions, potentially leading to the formation of metastable phases and incomplete transformations, further reducing the measured values. Impurities and the microstructural characteristics of an alloy also affect the DSC measurements but might not be accounted for in Thermo-Calc. The quality of thermodynamic databases and the methods used for data processing in DSC also contribute to these discrepancies. Despite these differences, as Table 3 illustrates, the latent heat values calculated using Thermo-Calc showed the same trend as the measured values. The best agreement between the measured latent heat values (via DSC) and the calculated values (via Thermo-Calc) was observed at lower cooling rates (R^2^ = 0.5). This finding was consistent with the understanding that the Thermo-Calc software package utilizes the Scheil–Gulliver model, which assumes rapid, non-equilibrium solidification. For alloys with lower silicon contents (5 and 7 wt.%), increasing the copper content from 1 to 2 wt.% leads to a lower latent heat release. However, further increases in copper content for these alloys result in a higher latent heat release. In contrast, for the AlSi9 alloy, any addition of copper reduces the amount of latent heat released. These findings align with previous discussions suggesting that alloy composition significantly impacts thermal properties during solidification, with silicon enhancing and copper reducing the latent heat released.

Thermo-Calc is widely used to calculate the latent heat of aluminum alloys, providing valuable insights into their solidification behavior and thermal properties. Despite neglecting the impact of cooling rates, this work has proven that Thermo-Calc can correctly assess the influence of alloy chemistry on latent heat in hypoeutectic AlSiCu alloys. By leveraging its advanced thermodynamic models, engineers and researchers can predict phase transformations and heat release during solidification.

## 4. Conclusions

Knowledge of the latent heat value for AlSiCu alloys is fundamental for the accurate simulation of casting processes. It impacts thermal modeling, microstructure prediction, quality control, and process optimization. By incorporating precise latent heat data into simulations, engineers can predict and control the behavior of an alloy during solidification, leading to improved product quality, reduced defects, and more efficient manufacturing processes.

The experimental data obtained from the DSC analysis of three distinct cooling/heating rates (6, 10, and 50 °C/min) for a specific range of AlSiCu alloys have been presented in this paper. Furthermore, the Thermo-Calc method has been applied to obtain theoretical results. This approach has provided the method for the prediction of the latent heat of the tested alloys, aluminum with wSi (5, 7, and 9%) and wCu (1, 2, and 4%).

Based on the presented data, the following can be concluded:
The increase in the silicon content from 5 to 9% shifted the liquidus temperatures to lower values by approximately 24 °C during both the heating and cooling cycles;The addition of copper, while maintaining a constant silicon content, lowered the liquidus temperature by approximately 20 °C during both the heating and cooling cycles;The increase in silicon content in the investigated alloys from 4.85% to 9.85% resulted in the increase in latent heat from 407.6 kJ/kg to 467.5 kJ/kg;The solidification paths were notably different for each cooling rate (Figure 5) and could affect the amount of released latent heat;Higher cooling rates, such as 50 °C/min, result in a reduced latent heat release compared to slower rates such as 10 °C/min and 6 °C/min.


The presented results will ultimately enhance the reliability and performance of the investigated aluminum alloys in various applications. It was found that the major alloying element, silicon, has a significant impact on the released latent heat. The observed results agree with the fact that a higher content of silicon means a greater fraction of primary silicon in the solidified structure, i.e., a higher value of latent heat, as well as a lower liquidus temperature.

## Figures and Tables

**Figure 1 materials-17-04228-f001:**
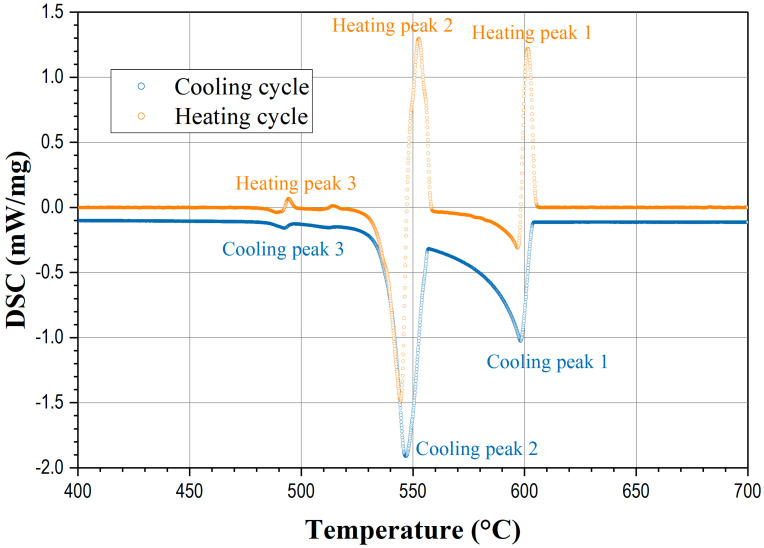
DSC cooling and heating curves for cooling/heating rate of 6 °C/min, with characteristic transformation temperatures, for alloy AlSi7Cu1.

**Figure 2 materials-17-04228-f002:**
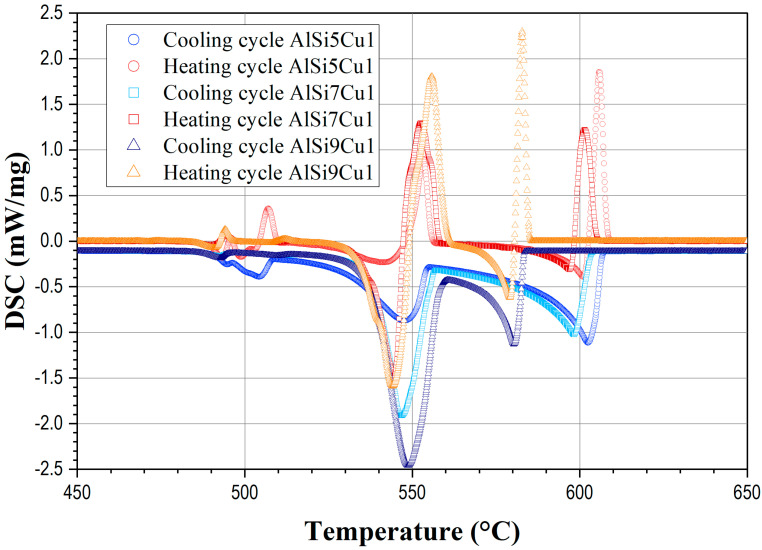
Characteristic transformation curves and corresponding transformation points for different silicon contents obtained by DSC at heating and cooling rates of 6 °C/min of AlSi (5, 7, and 9) Cu1 alloys.

**Figure 3 materials-17-04228-f003:**
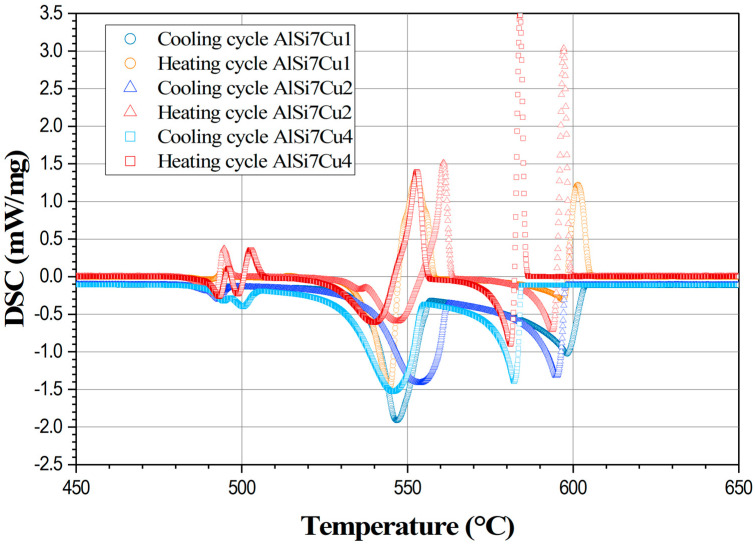
Characteristic transformation curves and corresponding transformation points for different copper contents obtained by DSC heating and cooling rates of 6 °C/min of AlSi7Cu (1, 2, and 4) alloys.

**Figure 4 materials-17-04228-f004:**
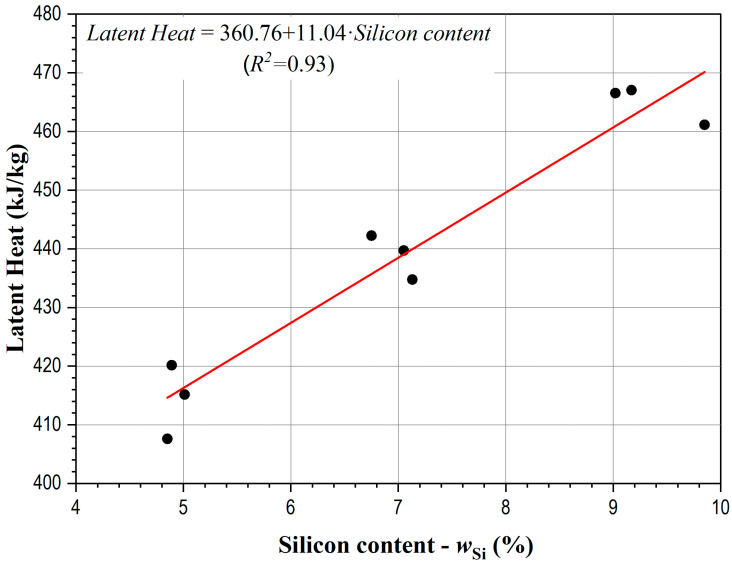
Impacts of various contents of silicon and copper on the latent heat of cast AlSiCu alloys at constant cooling rate of 6 °C/min.

**Figure 5 materials-17-04228-f005:**
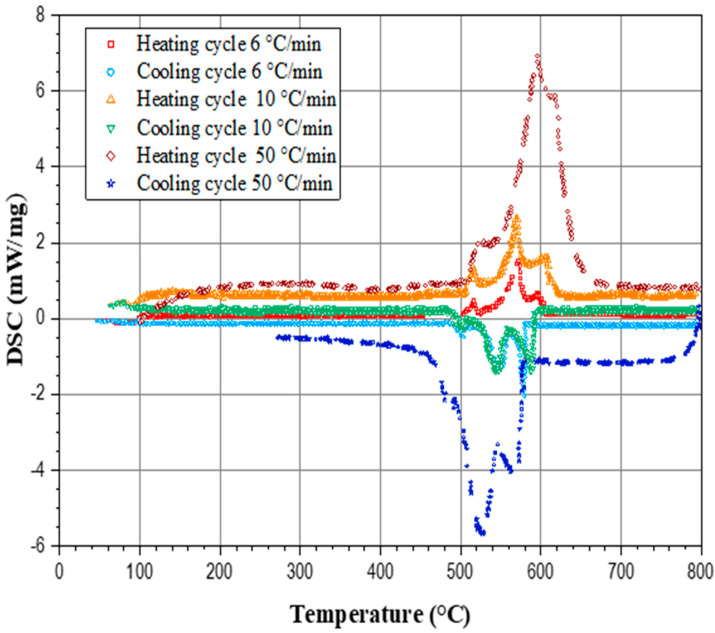
Heating (top) and cooling (bottom) DSC curves for the AlSi7Cu4 alloy, collected at various cooling and heating rates.

**Figure 6 materials-17-04228-f006:**
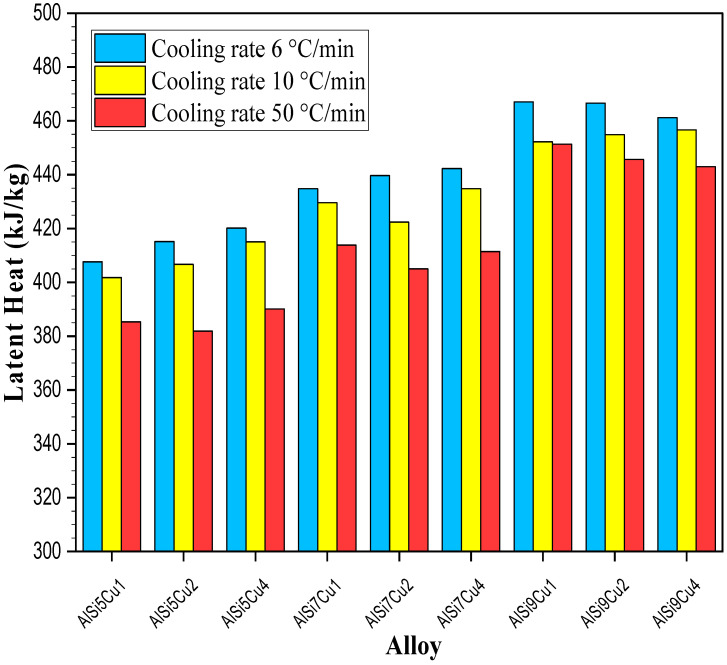
Impacts of various cooling rates on the amounts of latent heat released during solidification of investigated alloys.

**Table 1 materials-17-04228-t001:** Chemical compositions of investigated alloys (values are given in mass fraction percents).

Alloy	Si	Cu	Fe	Mg	Ti	Sr	Ni	Al
AlSI5Cu1	4.85	1.03	0.09	0.14	0.057	0.0009	0.009	93.82
AlSi5Cu2	5.01	2.06	0.1	0.15	0.062	0.0012	0.009	92.61
AlSi5Cu4	4.89	3.85	0.09	0.16	0.057	0.0035	0.009	90.94
AlSI7Cu1	7.13	0.96	0.12	0.28	0.098	0.0033	0.008	91.40
AlSi7Cu2	7.05	1.98	0.13	0.28	0.094	0.0027	0.009	90.45
AlSi7Cu4	6.75	4.38	0.12	0.29	0.091	0.0029	0.009	88.36
AlSI9Cu1	9.17	1.05	0.12	0.31	0.100	0.0042	0.007	89.24
AlSi9Cu2	9.02	2.44	0.12	0.31	0.096	0.0063	0.007	88.00
AlSi9Cu4	9.85	4.38	0.14	0.27	0.090	0.0035	0.009	85.26

**Table 2 materials-17-04228-t002:** The DSC thermal process cycles for tested AlSiCu aluminum alloys.

Temperature Segment	Temperature (°C)	Heating/Cooling Rate (°C/min)
Isothermal	25	0
Dynamic	800	6
Isothermal	800	0
Dynamic	25	6
Isothermal	25	0
Dynamic	800	10
Isothermal	800	0
Dynamic	25	10
Isothermal	25	0
Dynamic	800	50
Isothermal	800	0
Dynamic	25	50

**Table 3 materials-17-04228-t003:** The latent heat values for the series of AlSiCu alloys, measured using the DSC technique at various cooling and heating rates, are compared with latent heat values calculated using Thermo-Calc software.

Alloy	DSC 6 °C/min	DSC 10 °C/min	DSC 50 °C/min	Thermo-Calc
AlSi5Cu1	407.60	401.80	385.35	455.58
AlSi5Cu2	415.15	406.65	381.90	447.46
AlSi5Cu4	420.15	415.10	390.10	433.98
AlSi7Cu1	434.75	429.60	413.80	468.04
AlSi7Cu2	439.70	422.40	404.95	459.88
AlSi7Cu4	442.25	434.75	411.45	445.00
AlSi9Cu1	467.05	452.225	451.32	476.38
AlSi9Cu2	466.55	454.85	445.65	468.11
AlSi9Cu4	461.15	456.60	442.98	456.54

## Data Availability

Dataset available on request from the authors.

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
