# Peer review of "The Specificity of Determining the Latent Heat of Solidification of Cast Hypoeutectic AlSiCu Alloys Using the DSC Method"

_materials, 2024, doi:10.3390/ma17174228_

Round 1
Reviewer 1 Report
Comments and Suggestions for Authors
This manuscript presents useful data and interesting concepts. However, I have a few comments:
1. In the part Materials and Methods, you wrote that you used an aluminium crucible for the DSC analysis and tested at 800 °C. Is this correct? Aluminium crucible would melt.
2. Were samples repeatedly heated/cooled at various cooling/heating rates or were they separately analyzed? This is a little confusing from Table 2.
3. You describe the solidification of AlSi-eutectic as precipitation, the same also for other phases that solidify as precipitation. Please correct in whole manuscript.
4. In all Figs. presenting DSC vs. T mark exo and endo.
5. In Fig. 2 and 3 the curves overlap and can not be seen in detail what you describe in the manuscript. Please correct.
6. In Fig. 2 and 3 please mark how you read out T and L for the solidification and latent heat.
7. Fig. 3 it is written that data is presented for different silicon content, but it is presented regarding copper content. Please correct.
8. Fig.4: from where the presented data was obtained, from DSC curves? It is not clear. Enlarge the diagram to 10 silicon content.
9. Fig. 5: mark how the l was read. Are the marks in the legend correct? Please check as you describe, that at a slow cooling rate L should be the largest, but the curve is marked as the smallest.
10. The title of Table 3 is incorrect.
11. Line 281 you describe that the best agreement between DSC and TC results are at a higher cooling rate. This is not correct.
12. Line 285 you write that the higher copper content decreases the latent heat. This is not the case in table 3 as also from Fig.6.
13. Table 3: what are the conditions for the TC calculations, what is the cooling rate used?
14. Listing the references: please check, ie. 3, 13, ,,,,
Author Response
Dear Reviewer,
thank you very much for your valuable comments and invested time to improve our paper. We are sending our answers!
This manuscript presents useful data and interesting concepts. However, I have a few comments:
Q1.1. In the part Materials and Methods, you wrote that you used an aluminium crucible for the DSC analysis and tested at 800 °C. Is this correct? Aluminium crucible would melt.
A1.1. You are right, an alumina (Al2O3) crucible was used for the DSC experiments. The original text mistakenly stated, "aluminum crucible" instead of "alumina crucible". This error has been corrected.
Q1.2. Were samples repeatedly heated/cooled at various cooling/heating rates or were they separately analyzed? This is a little confusing from Table 2.
A1.2. Samples for DSC trials have been separately analyzed. It means for each cooling/heating rate, one sample has been selected and DSC experiments have been run.
Q1.3. You describe the solidification of AlSi-eutectic as precipitation, the same also for other phases that solidify as precipitation. Please correct in whole manuscript.
A1.3. Thank you for your proposal. We will consider it in our revised manuscript.
Q1.4. In all Figs. presenting DSC vs. T mark exo and endo.5.
A1.4. In DSC (Differential Scanning Calorimetry) curves, the distinction between exothermic and endothermic processes depends on the convention used, which can vary between instrument manufacturers and scientific fields. However, here is the most common interpretation:
Heating curve:
- Endothermic: Typically shown as a downward peak or dip
- Exothermic: Typically shown as an upward peak
Cooling curve:
- Endothermic: Typically shown as an upward peak
- Exothermic: Typically shown as a downward peak or dip
Accordingly, we have changed the manuscript adding the following text before Figure 1:
“In Figure 1, exothermic peaks are related to heating peaks 1, 2, and 3 (orange, pointing upward) while endothermic peaks are related to cooling peaks 1, 2, and 3 (blue, pointing downward). The heating cycle (orange) shows upward peaks, indicating heat is being released (exothermic). The cooling cycle (blue) shows downward peaks, indicating heat is being absorbed (endothermic).”
We have made this change as to avoid “overloading” the figures with additional text.
Q1.5.In Fig. 2 and 3 the curves overlap and can not be seen in detail what you describe in the manuscript. Please correct.
A1.5. Presented graphs are drawn based on original data obtained from the instrument. We have plotted all of them in a single figure trying to be efficient (one figure instead of three). If needed, we can split the plots in three figures.
Q1.6. In Fig. 2 and 3 please mark how you read out T and L for the solidification and latent heat.
A1.6. Peaks labeled as 1 in both heating and cooling cycles correspond to the precipitation of primary α-aluminum dendrites. We have not included specific transformation temperatures (T) in our manuscript because these vary significantly with the alloy's chemical composition and the applied cooling and heating rates. Figures 2 and 3 are presented to demonstrate the influence of silicon and copper content on the onset of precipitation for phases, rather than to indicate exact transformation temperatures. Latent heat (L) in DSC is calculated by integrating the area under the peak associated with a phase transition, relative to a defined baseline. This area is then converted to energy units using the instrument's calibration factor and normalized by the sample mass to give the specific latent heat (typically in J/g). The accuracy of this calculation depends on proper calibration, careful baseline selection, and precise determination of peak limits.
Q1.7. Fig. 3 it is written that data is presented for different silicon content, but it is presented regarding copper content. Please correct.
A1.7. You are right. We have corrected it.
Q1.8. Fig.4: from where the presented data was obtained, from DSC curves? It is not clear. Enlarge the diagram to 10 silicon content.
A1.8. All data presented in Figure 4 are obtained using DSC data related to calculated latent heat for various AlSiCu cast alloys (AlSi5Cu1, ALSi5Cu2, ALSi5Cu4, AlSi7Cu1, ALSi7Cu2, ALSi7Cu4, AlSi9Cu1, ALSi9Cu2, ALSi9Cu4) at cooling/heating rate of 6°C/min. Diagram has been enlarged to 10 wt.%Si.
Q1.9. Fig. 5: mark how the l was read. Are the marks in the legend correct? Please check as you describe, that at a slow cooling rate L should be the largest, but the curve is marked as the smallest.
A1.9. Please see Figure 6, which shows a clearer impact of cooling rates on the amount of released latent heat. In Figure 6, you can see that higher latent heat is related to slower cooling rates (6°C/min), while at higher cooling rates (50°C/min), the released latent heat for AlSi7Cu4 alloy is lower. This trend may not be easily visible in Figure 5.
Q1.10. The title of Table 3 is incorrect.
A.1.10. You are right, here is a new title for table 3. The latent heat values for the series of AlSiCu alloys, measured using the DSC technique at various cooling and heating rates, were compared with latent heat values calculated using Thermo-Calc software. It is changed in the manuscript.
Q1.11. Line 281 you describe that the best agreement between DSC and TC results are at a higher cooling rate. This is not correct.
A1.11. You are right. According to statistical analysis, the standard deviation between measured (DSC data) and calculated (TC) values is similar across cooling rates (at 6°C/min = 17.5, at 10°C/min = 17.3, and at 50°C/min = 19.5). This means the best agreement, according to statistical analysis, is at the lower cooling rate. This will be corrected in the manuscript.
Q1.12. Line 285 you write that the higher copper content decreases the latent heat. This is not the case in table 3 as also from Fig.6.
A1.12. You are correct. For alloys with lower silicon content (5 and 7 wt.%), increasing the copper content from 1 to 2 wt.% leads to lower latent heat release. However, further increases in copper content for these alloys result in higher latent heat release. In contrast, for the AlSi9 alloy, any addition of copper reduces the amount of latent heat released. This was changed in the manuscript.
Q1.13. Table 3: what are the conditions for the TC calculations, what is the cooling rate used?
A1.13. Thermo-Calc usually performs equilibrium calculations, which assume infinitely slow cooling rates. This approach considers that the system has enough time to reach thermodynamic equilibrium at each temperature.
Q1.14. Listing the references: please check, ie. 3, 13, ,,,,
A1.14. You are right, all references have been one more checked and where it was needed corrected.
Reviewer 2 Report
Comments and Suggestions for Authors
In the paper “Specificity of determining the latent heat of solidification of cast hypoeutectic AlSiCu alloys using the DSC method” the authors study the influence of chemical composition and cooling rates of the latent heat released during solidification. Many results are presented, but some aspects are unclear and need to be clarified.
I recommend a major revision. Next questions demand clarifications.
Table 1: Please insert a new column for Al rest
Line 121: The aluminum crucible can be used up to a temperature of 600 °C. Above this temperature it will melt. But the temperature used was 800 °C. Therefore, I think it's an error. You may have used alumina crucible. Please correct.
In Materials and Methods, Method, please insert cooling device (it is LN cooling?) and soft used (PROTEUS software?).
In my opinion, Table 2 should be deleted. It contains information that is explained in lines 125 to 128. Is no maintaining and cooling after last heating with heating rate = 50 °C/min. Also, cooling rates is -6, -10 and -50 C/min. It is also unclear if two consecutive cycles were performed using the same parameters or the experiments for each alloy, or were repeated twice. It would be helpful to insert a figure in which the thermal program to be presented.
Line 125:
I suggest replace the text:
Starting at room temperature (25 °C) the DSC’s samples were heated up to 800 °C held at this temperature for 10 minutes and cooled down to room temperature.
with:
Each thermal cycle included heating – maintaining - cooling process: 1: isothermal maintaining 10 minutes to 25 °C; 2. heating to 800 °C with three heating rates = 6, 10 and 50 °C/min; 3. isothermal maintaining 10 minutes to 800 °C; 4: cooling to 25 °C with three cooling rates = 6, 10 and 50 °C/min.
Line 142: Typical DSC curve…
Each peak must be explained, both on heating and cooling. All transformations are reversible? Also, some references are necessary. The explained peaks are specific to the cooling curve?
Notation of peaks with the same numbers for both heating and cooling is confusing. Heating is from left to right and cooling from right to left.
Line 147 Peak 1 correspond to the development of primary a-aluminum dendrites; Heating Peak 1 or cooling Peak 1? Or both? It is not clear.
Figure 1. DSC cooling and heating curves for cooling/heating rate of 6 °C/min, with characteristic transformation temperatures, for alloy AlSi7Cu1. The characteristic transformation temperatures are not presented. Same comment for Figure 2 and 3.
In my opinion, a table must be inserted in which characteristic transformation temperatures to be presented
Line 171: And in this case, we are talking about cooling curves. In my opinion the text most completed: Тhe shapes of the DSC cooling curves....
Line 186:….. at higher copper content (4 %), the appearance of two peaks in the DSC curves indicates the formation of two copper-rich phases. The same situation is shown in Figure 2 for the alloy AlSi7Cu1.
Line 229, 231 and 234: Insert references
3.2. Impact of various cooling rates on shape of DSC curves and calculated latent heats
The authors talk about the effect of the cooling rate on the microstructure. Did they do microscopic analyses? If not, references must be entered to confirm the explanations.
Line 289: Insert Table 3
Line 281: higher cooling rates... What it represents R2 = 0.5? The higher cooling rate is 50°C/min.
In my opinion, results must be rewritten.
Author Response
Dear Reviewer,
thank you very much for your invested time and valuable comments! We are sending our answers!
"I recommend a major revision. Next questions demand clarifications.
Q2.1: Table 1: Please insert a new column for Al rest
A2.1. A new column has been added
Q2.2. Line 121: The aluminum crucible can be used up to a temperature of 600 °C. Above this temperature it will melt. But the temperature used was 800 °C. Therefore, I think it's an error. You may have used alumina crucible. Please correct.
A2.2. An alumina (Al2O3) crucible was used for the DSC experiments. The original text mistakenly stated, "aluminum crucible" instead of "alumina crucible". This error has been corrected.
Q2.3. In Materials and Methods, Method, please insert cooling device (it is LN cooling?) and soft used (PROTEUS software?).
A2.3. The NETZSCH DSC 404 C Pegasus Differential Scanning Calorimeter typically uses a liquid nitrogen cooling system, which has been also applied during these experiments.
Q1.4. In my opinion, Table 2 should be deleted. It contains information that is explained in lines 125 to 128. Is no maintaining and cooling after last heating with heating rate = 50 °C/min. Also, cooling rates is -6, -10 and -50 C/min. It is also unclear if two consecutive cycles were performed using the same parameters or the experiments for each alloy, or were repeated twice. It would be helpful to insert a figure in which the thermal program to be presented.
A.2.4. For each alloy, the DSC experiment was conducted twice using identical parameters. The average latent heat value from these two tests was used for further analysis. We would like to retain the Table 2 in our manuscript. Retaining Table 2 enhances the manuscript's clarity and usability. While the information is summarized in the text, the table provides a quick visual reference with detailed parameters crucial for experiment replication and cross-study comparisons. Rather than deletion, any inaccuracies in the table should be corrected to maintain this valuable overview of experimental conditions.
Q2.5. Line 125:I suggest replace the text: Starting at room temperature (25 °C) the DSC’s samples were heated up to 800 °C held at this temperature for 10 minutes and cooled down to room temperature. with: Each thermal cycle included heating – maintaining – cooling process: 1: isothermal maintaining 10 minutes to 25 °C; 2. Heating to 800 °C with three heating rates = 6, 10 and 50 °C/min; 3. isothermal maintaining 10 minutes to 800 °C; 4: cooling to 25 °C with three cooling rates = 6, 10 and 50 °C/min.
A.2.5. We have accepted your suggestion and changed text according your proposal.
Q2.6. Line 142: Typical DSC curve…Each peak must be explained, both on heating and cooling. All transformations are reversible? Also, some references are necessary. The explained peaks are specific to the cooling curve? Notation of peaks with the same numbers for both heating and cooling is confusing. Heating is from left to right and cooling from right to left.
A1.6. All significant peaks for AlSiCu cast alloys have been explained in the text. Three distinct peaks are observed in both heating and cooling cycles of this AlSiCu series:
- Development of primary α-aluminum dendrites
- Precipitation of binary Al-Si eutectic phase
- Formation of minor copper-rich eutectic phases
Additional weak thermal events, such as magnesium and iron-rich phase precipitation, may also be detected depending on alloy composition. The heating cycle progresses from lower (400 °C) to higher temperatures (800 °C), while cooling follows the reverse. These peaks are well-established in the literature; however, we acknowledge that adding relevant references would strengthen the manuscript.
Q2.7. Line 147 Peak 1 correspond to the development of primary a-aluminum dendrites; Heating Peak 1 or cooling Peak 1? Or both? It is not clear. Figure 1. DSC cooling and heating curves for cooling/ heating rate of 6 °C/min, with characteristic transformation temperatures, for alloy AlSi7Cu1. The characteristic transformation temperatures are not presented. Same comment for Figure 2 and 3. In my opinion, a table must be inserted in which characteristic transformation temperatures to be presented
A2.7. Peaks labeled as 1 in both heating and cooling cycles correspond to the precipitation of primary α-aluminum dendrites. We have not included specific transformation temperatures in our manuscript because these vary significantly with the alloy's chemical composition and the applied cooling and heating rates. Figures 2 and 3 are presented to demonstrate the influence of silicon and copper content on the onset of precipitation for phases, rather than to indicate exact transformation temperatures.
Q2.8. Line 171: And in this case, we are talking about cooling curves. In my opinion the text most completed: Тhe shapes of the DSC cooling curves....
A2.8. Accepted. We have changed this according to your proposal.
Q2.9. Line 186:….. at higher copper content (4 %), the appearance of two peaks in the DSC curves indicates the formation of two copperrich phases. The same situation is shown in Figure 2 for the alloy AlSi7Cu1.
A2.9. You are correct in your observation. In alloys with lower copper content, two peaks can be faintly discerned. However, in alloys with higher copper content, these peaks are significantly more pronounced. The intensity of these peaks correlates directly with the copper concentration in the alloy.
Q2.10. Line 229, 231 and 234: Insert references 3.2. Impact of various cooling rates on shape of DSC curves and calculated latent heats The authors talk about the effect of the cooling rate on the microstructure. Did they do microscopic analyses? If not, references must be entered to confirm the explanations.
A2.10. This paper aims to comprehensively assess the accuracy of calculated latent heat for a specific range of cast AlSiCu alloys under various solidification conditions. Our investigation focuses on the effects of silicon (5 % to 9 %) and copper (1 % to 4 %) concentrations on the latent heat released during solidification. To analyze the impact of cooling rates, we applied three distinct rates: 6, 10, and 50 °C/min in each test. We combined experimental tests using Differential Scanning Calorimetry with theoretical results obtained through the Thermo-Calc method. This approach provides a method for predicting the latent heat of the tested alloys. Microscopic analysis of the specimens was not included in this study.
Q2.11. Line 289: Insert Table 3
A2.11. Accepted, changed.
Q2.12. Line 281: higher cooling rates... What it represents R = 0.5? The higher cooling rate is 50 °C/min.
A2.12. The R² value of 0.5 in this context refers to the coefficient of determination, which is a statistical measure used to assess how well a regression model fits the data from Table 3.
Reviewer 3 Report
Comments and Suggestions for Authors
The study and analysis of patterns of change and modeling of thermodynamic properties of metal melts are urgent tasks of modern physical chemistry. About this evidenced by the growing number of publications in leading periodicals devoted to this topic, a large number of international scientific forums aimed at solving fundamental and applied problems related to this field of knowledge. In this study, the authors of the article used the DSC method for a comprehensive assessment of the accuracy of the calculated latent heat for a certain range of cast AlSiCu alloys, taking into account their solidification under different cooling conditions. The tests included different concentrations of the two most important alloying elements: The obtained results are valuable for science and practice, but there are several questions:
1. Is there a correlation of excess heat capacity of melts with enthalpies of mixing, can they be described by empirical equations?
2. How to simulate metastable transformations with the participation of a supercooled liquid phase: delamination of melts and related transformations? Is it possible within the framework of the proposed approach?
3. In the conclusions of the article, I propose to show its scientific value.
Author Response
Dear Reviewer,
thank you very much for your invested time and valuable coments! Now We are sending our answers!
Q3.1. Is there a correlation of excess heat capacity of melts with enthalpies of mixing, can they be described by empirical equations?
A3.1. Excess heat capacity of melts and enthalpies of mixing are often correlated due to their shared origin in non-ideal component interactions. This relationship can be described using empirical equations, such as the Redlich-Kister polynomial expansion or models derived from the CALPHAD method. These equations typically account for composition and temperature dependence. While useful for many systems, the accuracy of these empirical models can vary depending on the specific alloy and the range of conditions considered.
Q3.2. How to simulate metastable transformations with the participation of a supercooled liquid phase: delamination of melts and related transformations? Is it possible within the framework of the proposed approach?
A3.2. Simulating metastable transformations with supercooled liquid phases, including melt delamination, requires advanced computational methods that go beyond equilibrium thermodynamics. These simulations often combine thermodynamic models with kinetic considerations, using techniques like phase-field modeling or molecular dynamics. Such approaches can account for supercooling effects and interfacial phenomena. The feasibility within a proposed framework depends on its ability to handle non-equilibrium conditions and incorporate relevant physical parameters. We did not consider in our paper to simulate metastable transformations with the precipitation of a supercooled liquid phase.
Q3.3. In the conclusions of the article, I propose to show its scientific value.
A3.3. Knowledge of latent heat values for AlSiCu alloys is crucial for advancing materials science and engineering. It enables more accurate alloy design, improves understanding of phase transformations, and enhances thermal management strategies. This data also strengthens computational models and contributes to fundamental thermodynamics research. Ultimately, precise latent heat information leads to improved product quality, more efficient manufacturing processes, and the development of innovative materials with tailored properties.
Round 2
Reviewer 1 Report
Comments and Suggestions for Authors
Remarks are made in the manuscript.

Author Response
Dear Reviewer,
Thank you very much for your invested time and important questions and remarks!
Answers to reviewers related to Ref.: Ms. No. materials-3158569, Article: Specificity of determining the latent heat of solidification of cast hypoeutectic AlSiCu alloys using the DSC method, Authors: Mile Đurđević , Vladimir Jovanovic , Mirko Komatina , Srecko Stopic
We have grouped the answers to pages as commented by reviewer except for the multiple comment/question “Poor visibility/resolution between the peacks. How the energies and T were read out?” which is answered once (at first comment).
Page 4 precipitation
A1.1 We have changed word “precipitation” with “appearance” where highlighted.
Page 6 Poor visibility/resolution between the peacks. How the energies and T were read out? (in title of Figure 2)
A1.2 The energy is measured as the heat flow rate (in milliwatts per milligram, mW/mg) on the y-axis of the DSC curve, where the peak area corresponds to the latent heat or enthalpy change (ΔH) associated with the phase transformation or solidification of the AlSiCu alloy. The temperature is measured on the x-axis of the DSC curve, and the characteristic temperatures such as the onset, peak, and endpoint temperatures are used to identify the specific phase transitions or solidification events occurring in the alloy.
This answer apply to same comment on page 7 (in title of Figure 3)
Page 7 Lines 188 and 196 precipitation temperature, Line 200 precipitate
A1.3. We have changed words “precipitation temperature” with “nucleation temperature” and “precipitated” with “cristallized” where highlighted.
Page 8 Line 208, Line 220 precipitation, Line 221 precipitate
A1.4 We have added text “, read from DSC curves,” to be more precise, as suggested by the reviewer.
A1.5 We have changed words “precipitation temperature” with “nucleation temperature” and “precipitated” to “cristallized” (line 200) where highlighted.
Page 9 Line 236 (title of Figure 5)
A1.6 You are right, we also noticed it. Here is our potential explanation for this anomaly. The apparent discrepancy between peak heights in the DSC curves and calculated latent heat values can be explained by considering that latent heat is determined by the area under the curve, not just peak height. Slower cooling rates allow for more complete phase transformations over a longer time, potentially resulting in broader peaks with larger total areas. This can lead to higher overall latent heat release, even if the instantaneous rate (peak height) is lower. Additionally, slower cooling may promote the formation of microstructures that release more total latent heat, despite less intense momentary heat flow.
Page 11 Line 296 lower latent … Line 306 (title of Table 3)
A1.7 The latent heat of solidification increases significantly with higher silicon content in aluminum alloys, regardless of cooling rates. The addition of copper to these alloys has a more nuanced effect on latent heat, depending on the silicon content and cooling rate. For AlSi5 alloys, copper addition increases latent heat across all investigated cooling rates. In AlSi7 alloys, copper addition increases latent heat at slower cooling rates (5°C/min) but decreases it at higher cooling rates. For AlSi9 alloys, the effect of copper is more variable. At 6°C/min and 60°C/min cooling rates, copper addition reduces latent heat release, while at 10°C/min cooling rate, copper slightly increases the latent heat released. These findings demonstrate that silicon content consistently enhances latent heat release during solidification, while copper's influence is more complex and dependent on both silicon content and cooling rate. The results align with previous research indicating that alloy composition significantly impacts thermal properties during solidification, with silicon playing a particularly influential role in determining the amount of latent heat released.
A1.8 Thermo-Calc software package utilizes the Scheil-Gulliver model, which assumes rapid, non-equilibrium solidification
We have included our answers in our new version! I hope that this improved version can be accepted for publishing in Journal Materials!
Reviewer 2 Report
Comments and Suggestions for Authors
Line 125-126: Please delete Starting at room temperature (25 °C) the DSC’s samples were heated up to 800 °C held at this temperature for 10 minutes and cooled down to room temperature. with:
Table 2. If you decide to keep Table 2, please correct it. Isothermal 25 and Dynamic step for heating 50 (°C) are missing. The cooling rate should is not with ”-”.
A corrected version is presented below.
Temperature Segment |
Temperature (°C) |
Heating/Cooling Rate (°C/min) |
Isothermal |
25 |
0 |
Dynamic |
800 |
6 |
Isothermal |
800 |
0 |
Dynamic |
25 |
6 |
|
|
|
Isothermal |
25 |
0 |
Dynamic |
800 |
10 |
Isothermal |
800 |
0 |
Dynamic |
25 |
10 |
|
|
|
Isothermal |
25 |
0 |
Dynamic |
800 |
50 |
Isothermal |
800 |
0 |
Dynamic |
25 |
50 |
Line 148: … DSC trace…
According with ICTAC nomenclature of thermal analysis (IUPAC Recommendations 2014) - DOI 10.1515/pac-2012-0609 Pure Appl. Chem. 2014; 86(4): 545–553, Terminology, the accepted term is thermal analysis curve, for a specific technique, in this case DSC, the accepted term is DSC curve.
I suggest replace DSC trace with DSC curve.
Author Response
Dear Reviewer,
thank you very much for your invested time and very important comments!
We have accepted all your suggestions and changed the text accordingly.
Attached you can see this version.
I hope, that this improved version can be accepted in current form!
